# Alternative Rooting Methods for Medicinal Cannabis Cultivation in Denmark—Preliminary Results

**DOI:** 10.3390/plants12112216

**Published:** 2023-06-03

**Authors:** Bruno Trevenzoli Favero, Jacob Kromann Salomonsen, Henrik Lütken

**Affiliations:** Crop Sciences Section, Department of Plant and Environmental Sciences, Faculty of Science, University of Copenhagen, 2630 Tåstrup, Denmarkhlm@plen.ku.dk (H.L.)

**Keywords:** *C. sativa*, chimeric, *Rhizobium rhizogenes*, root, rooting phenotype, WinRhizo

## Abstract

*Cannabis sativa* L. is typically propagated through micropropagation or vegetative cuttings, but the use of root-inducing hormones, such as indole-3-butyric acid (IBA), is not allowed for growing medicinal cannabis in Denmark. This study examined alternative rooting treatments, including *Rhizobium rhizogenes* inoculation, water-only as well as IBA treatments, in eight cannabis cultivars. PCR on root tissue suggested that 19% of *R. rhizogenes-*inoculated cuttings were transformed. These were derived from “Herijuana”, “Wild Thailand”, “Motherlode Kush”, and “Bruce Banner”, indicating a variation in cultivar susceptibility toward *R. rhizogenes*. A 100% rooting success was achieved regardless of cultivar and treatment, suggesting that alternative rooting agents are not required for efficient vegetative propagation. However, rooted cuttings differed in shoot morphology with improved shoot growth in cuttings treated with *R. rhizogenes* (195 ± 7 mm) or water (185 ± 7 mm) while inhibited shoot growth under IBA treatment (123 ± 6 mm). This could have advantageous economic implications should cuttings not treated with hormone reach maturity faster than those exposed to the hormone, thereby contributing to completing a full growing cycle more effectively. IBA exposure increased root length, root dry weight, and root/shoot dry weight ratio compared to cuttings treated with *R. rhizogenes* or water but simultaneously inhibited shoot growth compared to these.

## 1. Introduction

*Cannabis sativa* L. belongs to the family Cannabaceae, and it is among the first plants to be cultivated. Its uses include fiber, fuel, food, cosmetics, and medicine [1]. Drug-type cannabis contains a range of bioactive compounds, including cannabinoids, terpenoids, and alkaloids [2]. In January 2018, the program on medicinal cannabis was initiated in Denmark, making it possible for doctors to prescribe cannabis-based medicine to patients with unresolved medical conditions [3], thus fostering research locally [4]. Additionally, a license to cultivate medicinal cannabis has to be applied for, as selling dried cannabis inflorescences/flowers (cannabis bulk) is only allowed upon obtaining official approval for consistently producing high-quality inflorescences, which are measured as high, uniform levels of cannabinoids. Eleven companies have so far been given the final approval to sell bulk cannabis [5], and the trial was recently extended, making it permanently legal for companies to grow medicinal cannabis in Denmark [6]. Today’s medicinal cannabis production in Denmark is highly specialized and technical but simultaneously embedded in a framework that allows for limited use of some modern agricultural practices, for example, within vegetative propagation. In general, cannabis can be propagated through seed [7], vegetative cuttings [8], or in vitro propagation [9,10]. Propagation through vegetative cuttings is likely the most realistic practice in the Danish context, as it ensures consistent cannabinoid levels and similar growth patterns between plants. Caplan [11] evaluated the effects of several treatments (leaf tip removal, leaf number, position of cuttings, and rooting hormone) on the rooting success of cannabis cuttings. These authors recommend having at least three fully expanded leaves on cuttings, and the position of cuttings on mother plants had no effect on rooting success [11]. Knowledge of such propagation practices is important from a practical perspective since it is manual labor and, therefore, associated with considerable costs. In order to initiate rooting, cuttings are commonly exposed to auxins, a group of naturally occurring hormones that are important regulators in plants regarding the division, expansion, and differentiation of cells as reviewed by Enders et al. [12]. While such products are not explicitly mentioned in the official documents regarding Danish medicinal cannabis production [13,14], their prohibition has been confirmed [15]. Growers may then choose to import already rooted cuttings from elsewhere. Therefore, novel methods to induce root formation in cuttings of drug-type cannabis are needed in Denmark. Alternative rooting products for cannabis based on biological compounds are available on the market, but few have been tested scientifically. A study by Caplan [11] tested a willow (*Salix alba*) bark extract gel against an indole butyric acid (IBA) gel on drug-type cannabis cuttings with reported rooting successes in 40% and 84%, respectively. Willow bark extracts contain the phytohormone salicylic acid, which has been associated with improved root growth in plant cuttings, for example, in soybean [16]. Dialogue with Danish cannabis growers indicates that biostimulants, e.g., protein hydrolysates and algae extracts, are applied as alternative root agents, although with variable success. The current study focused on the root-inducing (Ri) ability that the rhizogenic bacterium *Rhizobium rhizogenes* possesses. This is a naturally occurring soil bacterium that exerts a pathogenic lifestyle conferred by the root-inducing plasmid (pRi) that strains within *R. rhizogenes* [17] and *R. radiobacter* as reviewed by Desmet et al. [18]). The transfer DNA (T-DNA) resembles the part of the plasmid that can horizontally transfer to the host plant genome and cause “hairy roots” to form at the infection site, which is the main symptom of successful T-DNA integration in the host [19,20]. Agropine strains of *R. rhizogenes* contain two separate T-DNA regions, namely, the left (T_L_) and the right (T_R_). T_L_ harbors 18 open reading frames (ORFs), including the *root oncogenic loci* (*rol*) genes *rol*A, *rol*B, *rol*C, and *rol*D, which are essential to initiate hairy root formation, whereas T_R_ contains a *rol*B homolog and two genes (*aux1* and *aux2*) that are involved in auxin synthesis [21].

It is sometimes possible to regenerate plants from transformed tissue, resulting in complete, transgenic plants recognized as the hairy root or Ri phenotype [18]. Such plants are considered naturally transformed, and this phenomenon has been reported to occur in nature, e.g., tobacco (*Nicotiana glauca*) [22], cultivated sweet potato (*Ipomoea batatas*) [23], and *Linaria* [24]. Using naturally occurring wild-type bacterial strains is, therefore, considered a non-GMO (genetically modified organism) method in the European Union so long as they remain unmodified [25]. Several general traits that often follow a natural transformation include dwarfed stature, wrinkled leaves, increased rooting ability, altered flowering, and reduced fertility (reviewed by Cassanova et al. [26] and Desmet et al. [18]). Additionally, transformed tissues can produce elevated levels of medically important secondary metabolites, e.g., anthraquinones, saponins, flavonoids, alkaloids, anthocyannins, and terpenes [27,28]. A rooting deficiency of almond (*Prunus dulcis*) cuttings of the cultivar “Supernova” was overcome after transformation with wild-type *R. rhizogenes* [29]. In order to fully realize such potential for medicinal cannabis, transformation and regeneration would be necessary, but only the former has been demonstrated in *C. sativa* by Feeney and Punja [30] and Wahby et al. [31]. Thus, natural transformation can potentially serve as an important tool in medicinal cannabis, effectively bypassing the legal issues that genetic engineering implies. Agronomically useful traits regarding rooting can also be obtained without altering shoot properties. Lambert and Tepfer [32] investigated chimeric apple cuttings (transgenic roots, WT aerial part) to overcome a rooting deficiency, which was achieved on cuttings of thee-year-old trees (rootstock M9b) after inoculation with *R. rhizogenes* (A4 strain). In the current study, we investigated rooting performance in chimeric cannabis cuttings in a similar manner to Lambert and Tepfer [32]. We specifically aimed to investigate whether *R. rhizogenes* inoculation could improve rooting in cannabis cuttings versus IBA or treated solely with water. 

## 2. Results

In the current study, the rooting performance of cannabis cuttings was compared between three treatment groups in two repetitions displaced in time. The rooting success, meaning whether there was visible root(s) after 25 days of growth or not, was recorded for each cutting and used to evaluate whether all data across repetitions and cultivars could be combined (Table 1). The highest rooting success within treatments for both repetitions was that of IBA (98 ± 3%). This was followed by a 95 ± 5% and 86 ± 14% success rate for the *Rhizobium rhizogenes* (A4) and H_2_O treatment, respectively (Table 1). In repetition 1, 83 out of 96 cuttings rooted (86%), whereas all cuttings of repetition 2 rooted successfully. Rooting failure was most pronounced in the H_2_O treatment, where 25% of the “Bruce Banner” and 0% of the “Motherlode Kush” cuttings rooted (Table 1). As rooting averages differed between repetitions, it was hypothesized that these differences were more related to repetitions rather than cultivars. Therefore, the data was further examined before being combined or separated. A chi-squared test between cultivars in both repetitions was significant (*p* ≤ 0.0102), indicating a difference in rooting success among them (data not shown). The poor rooting success of “Bruce Banner” (25%) and “Motherlode Kush” (0%) in repetition 1 was removed from the data before running the test again. This rendered the expected rooting success in alignment with the observed frequencies, ultimately providing the basis for combining data across cultivars (*p* ≤ 0.2786).

Finally, the rooting success between repetitions was analyzed, excluding “Bruce Banner” and “Motherlode Kush” cuttings from repetition 1, which was significantly different (*p* ≤ 0.0093), suggesting that data across repetitions should not be combined. Therefore, this analysis provided the basis for combining data across cultivars but not across repetitions.

### 2.1. Biometric Analyses—Repetition 1

Upon termination of this experiment, the length of shoots of A4 cuttings measured 91 ± 5 mm, which was significantly higher (*p* ≤ 0.0006) than IBA (65 ± 4 mm) and H_2_O (76 ± 6 mm) cuttings (Figure 1a). The root/shoot dry weight ratio showed growth distribution between root and shoot, which was significantly higher (0.22 ± 0.01) for IBA-exposed cuttings compared to A4 (0.14 ± 0.001) and H_2_O (0.13 ± 0.02) cuttings (*p* ≤ 0.0001) (Figure 1c). This indicated that cuttings exposed to IBA allocated more resources to root growth relative to shoot growth compared to cuttings of the A4 and H_2_O treatment. This is further exhibited by root length measures that show a significant difference between IBA (667 ± 75 cm) and H_2_O (337 ± 86 cm) cuttings (*p* ≤ 0.0063) (Figure 1d). No statistically significant difference was found for the root length of A4 (456 ± 56 cm) cuttings compared to IBA and H_2_O (Figure 1d). Additionally, there was no significant difference among treatment groups for the parameters shoot dry weight (Figure 1b), root dry weight (Figure 1e), and root diameter (Figure 1f). 

### 2.2. Biometric Analyses—Repetition 2

After terminating the second repetition, the shoot length of A4 (195 ± 7 mm) and H_2_O (185 ± 7 mm) cuttings both measured significantly longer than IBA cuttings (123 ± 6 mm) at *p* ≤ 0.0001 (Figure 2a). Similar results were observed in the shoot growth parameter, which meant the difference in shoot length measured at the time of taking the cutting and at the experimental termination (Figure 2g). A4 and H_2_O-treated cuttings grew significantly longer (114 ± 7 mm and 103 ± 7 mm, respectively) than IBA cuttings (43 ± 5 mm), which was significant at *p* ≤ 0.0001 for both comparisons (Figure 2g).

The root/shoot dry weight ratio illustrated an altered growth pattern for cuttings exposed to IBA through a significantly increased proportion of biomass found in roots rather than shoots (0.32 ± 0.02) when compared to A4 (0.11 ± 0.00) and H_2_O (0.12 ± 0.00) cuttings (*p* ≤ 0.0001) (Figure 2c). A4 and H_2_O were not statistically different, indicating that inoculation with *R. rhizogenes* did not affect the growth of cuttings in a similar manner as IBA (Figure 3a–c).

The root was significantly longer (*p* ≤ 0.001) for IBA cuttings (2331 ± 233 cm) compared to A4 (1405 ± 143 cm) and H_2_O cuttings (1533 ± 170 cm), and no statistical difference was observed between the A4 and H_2_O treatment (Figure 2d). Similarly, root dry weight was highest for IBA-exposed cuttings (226 ± 21 mg) and significantly so (*p* ≤ 0.0001) tested against the lengths for A4 (94 ± 9 mg) and H_2_O-treated (121 ± 14 mg) cuttings (Figure 2e). A significant difference was found between the root diameter of IBA (0.39 ± 0.00 mm) and A4 cuttings (0.36 ± 0.00 mm) but not for either of these treatments compared to H_2_O cutting diameters (0.37 ± 0.00 mm). In repetition 2, there were no significant differences among treatment groups in the shoot dry weight averages (Figure 2b). 

### 2.3. Detection of rolB, virD, and EF1α Gene Fragments in Root Tissue of Cuttings

Root tissue samples of 32 A4 cuttings, 1 IBA cutting, 1 H_2_O cutting, as well as nutrient solution samples from each treatment were analyzed by PCR in repetition 2 of this experiment (Figure 4). This was primarily conducted to verify whether cuttings had been successfully infected by *R. rhizogenes,* but nutrient solution samples were included to check for the spread of *R. rhizogenes* between aeroponic growth units, which were positioned immediately adjacent to one another. The criteria for inferring the transfer of T_L_-DNA were the presence of *rol*B and the absence of *vir*D, which was located outside the T-DNA regions of the plasmid; thus, it should not be present in the case of successful transformation. The reference gene *EF1α* was present in all cuttings that were tested, which verifies the adequate quality of the extracted DNA (Figure 4). Bands for *rol*B were found in 21 of 34 cuttings (62%) (Figure 4). Plasmid remnants indicated by *vir*D bands were seen in 32% of the cuttings (Figure 4; samples 2, 3, 7, 9, 10, 11, 14, 17, 23, 34, 35), all of which were of the A4 treatment group. The *rol*B bands were not present in the nutrient solutions (Figure 4; samples 200, 201, 202) as well as in root tissue from an IBA- and an H_2_O cutting; similarly, no bands for *vir*D were observed (Figure 4; samples 46, 81).

## 3. Discussion

The aim of this work was to investigate *R. rhizogenes* as an alternative rooting agent to IBA in drug-type cannabis cuttings. Quantitative biometrics were presented to analyze differences in rooting abilities between treatments. A key finding was that *R. rhizogenes* (A4) and H_2_-treated cuttings proved to have longer and well-established shoots, yet with shorter roots than IBA-treated cuttings. Moreover, results from repetition 1 showed improved rooting abilities of IBA-exposed cuttings compared to A4 and H_2_O cuttings, as displayed by the significantly higher root/shoot dry weight ratio of IBA cuttings (0.22 ± 0.01) compared to A4 (0.14 ± 0.001) and H_2_O (0.13 ± 0.02) cuttings (Figure 1c). This was also the case in repetition 2 with root/shoot dry weight ratios, measuring 0.32 ± 0.02 (IBA), 0.11 ± 0.00 (A4), and 0.12 ± 0.00 (H_2_O) (Figure 2c). Altered proportions of biomass in roots and shoots after IBA exposure was also found by Hunt et al. [33] in a study on vegetative pine cuttings (*Pinus elliottii* var. *elliottii* × *P. caribaea* var. *hondurensis*). These were either dipped in 16.000 mg L^−1^ IBA for 5 s or water as a control and after 13 weeks, root/shoot dry weight ratios were significantly different for clones with IBA and clones without. In another study, beech (*Fagus sylvatica*) seedlings exposed to >1000 mg L^−1^ IBA had significantly greater root/shoot dry weight ratios (1.28) than untreated controls 105 days after exposure [34].

In repetition 2, the root and shoot growth measurements generally showed similar results for A4 and H_2_O treatment groups and several simultaneous significant differences compared to IBA (Figure 2a,c–e,g). Cuttings exposed to IBA exhibited increased rooting (Figure 2c–e) as well as decreased shoot growth (Figure 2a,g). Average total root lengths for A4 and H_2_O cuttings were not statistically significant in both repetitions of this experiment (Figure 1d and Figure 2d), suggesting no improved rooting abilities of cannabis cuttings inoculated with *R. rhizogenes*. Comparable results were reported by Hatta [35] on softwood cuttings of two jujube (*Ziziphus jujuba*) cultivars (“Contorta” and “Li”). Ten weeks after inoculation with strains A4 and TR105 of *R. rhizogenes*, neither strain had an effect on the length of the longest root compared to water controls, although the longest roots of “Li” cuttings were significantly longer than those of “Concorta” cuttings (*p* ≤ 0.05), suggesting different innate rooting abilities between cultivars [35]. A similar cultivar difference was found when measuring the number of roots per cutting, which also revealed significantly more roots (*p* ≤ 0.05) in cuttings inoculated with the TR105 strains, whereas the A4 strain yielded no more roots on average than water controls [35]. Increased total root lengths (*p* ≤ 0.01) and root dry weights (*p* ≤ 0.05) have been shown in mint (*Mentha piperita*) cuttings upon inoculation with *R. rhizogenes* (strain A16), although root dry weights were increased more by inoculation with *Bacillus megatorium* (strain M3) than *R. rhizogenes*. In the present work, root dry weights of A4 cuttings were the same as those of H_2_O cuttings in both repetitions (Figure 1d and Figure 2d).

The effect of IBA exposure on shoot growth in cannabis cuttings has apparently not yet been illustrated in the scientific literature. In this work, A4 and H_2_O-treated cuttings grew significantly longer during 25 days of aeroponic propagation (114 ± 7 mm and 103 ± 7 mm) than IBA cuttings (43 ± 5 mm). Longer shoots are preferred by the industry as mature plants can be obtained earlier. The non-significant relationship between A4 and H_2_O cutting shoot growth indicated a growth retarding effect of IBA rather than improved shoot growth in A4 cuttings. This could have advantageous economic implications should cuttings not treated with hormone reach maturity faster than those exposed to the hormone, thereby contributing to completing a full growing cycle more effectively. Complete immersion of woody ornamental cuttings in IBA solutions is known to cause shoot growth retardation. Initial growth alterations upon different hormone exposures may, however, be compensated at a later growth stage, eventually yielding similar marketable plants (reviewed by Blythe et al. [36]). In beech, cuttings exposed to 250–1000 mg L^−1^ IBA for 10 min showed increased shoot lengths after 54 days, while cuttings exposed to 2000 mg L^−1^ IBA were no different from cuttings without hormone, possibly indicating a threshold value of IBA concentration, duration of the exposure, or a combination of both [34]. Consequently, these authors concluded that the optimum duration of IBA exposure in beech cuttings depended on IBA concentration [34]. Cannabis cuttings in this experiment were exposed to a 100 mg L^−1^ IBA solution following the “long basal soak” protocol for herbaceous cuttings by Kroin [37], which provided guidelines for hormone solution concentrations and several ways of exposing plant material. This method was also employed by Gehlot [38] on vegetative propagation of neem (*Azadirachta indica*), although the actual exposure time was not listed. Nevertheless, across a range of hormone solution concentrations (250; 500; 750; 1000; 1500 mg L^−1^), 250 mg L^−1^ IBA provided the best results, e.g., the highest rooting success (80%) and highest number of roots (71), compared to other hormone concentrations [38]. In the current study, Kroin’s [37] “long basal soak” method was chosen over more time-efficient methods because it was suspected that bacterial infection of cuttings would be less likely to occur in seconds or minutes of inoculation. 

In order to assess whether *R. rhizogenes* improved the rooting performance of cannabis cuttings, it was essential to determine if inoculation had led to the infection of the cuttings. A total of 12 out of 32 (38%) cuttings tested showed bands for *EF1α* and *rol*B but not for *vir*D, thus meeting the criteria for inferring that an infection, leading to hairy root formation, had occurred (Figure 4). It could, perhaps, be beneficial to evaluate the infection success over gradients of bacterial concentrations or durations of the inoculation step. In *Arabidopsis thaliana*, transformation with *Agrobacterium tumefaciens* (strain GV3101(pmP90)) has been attempted at a range of bacterial concentrations (OD_600_ ≤ 0.15–1.75), with little reported effects on transformation rates (0.21 ± 0.05–0.57 ± 0.15%) [39]. Bacterial cultures used for that experiment were grown to the stationary phase (ca. 18–24 h growth) and subsequently diluted to different concentrations, and an additional late stationary phase culture (84 h growth; adjusted to OD_600_ ≤ 0.8) transformed as efficiently (0.50 ± 0.05%) as the younger cultures [39]. 

The current work suggests that cultivars “Herijuana”, “Bruce Banner”, “Motherlode Kush”, and “Wild Thailand” can be infected by *R. rhizogenes,* whereas “Hindu Kush”, “California Orange”, “The Pure”, and “Big Bud” are less susceptible under the reported conditions (Figure 4). *Agrobacterium*-mediated transformation of *C. sativa* was first shown using *A. tumefaciens* in tissue culture by Feeney and Punja [30] on four fiber type cultivars “Uniko-B”, “Kompolti”, “Anka”, and “Felina-34”. Additional fiber types cultivars “Futura77”, “Delta-405”, and “Delta-Llosa” and drug-type cultivars “CAN0111” and “CAN0221” were successfully transformed with *R. rhizogenes* (strains A4 and AR10) with varying, though insignificant, successes, ranging from 43–98% [31]. Additionally, “CAN0221” was inoculated with eight wild-type *R. rhizogenes* strains, which were all able to induce hairy roots on seedlings without statistically detectable differences in susceptibility toward strains [31]. It should be mentioned that despite being susceptible to *R. rhizogenes* infection, differences in response to infection occur between cultivars in vitro, for example, by frequency of root induction or the number of roots [40]. 

The rooting success of *C. sativa* cuttings was used to assess whether all biometric data between repetitions 1 and 2 could be combined. A series of Chi-squared tests provided the basis for combining data across cultivars but not repetitions, suggesting that the conditions between repetitions differed too much to analyze all data collectively (data not shown). In the aeroponic growth units used in this study, cuttings were positioned in close proximity in a fixed grid pattern (≤10 cm between stems of cuttings). As a result, cuttings measuring 130–150 mm in length would most likely benefit from further trimming of fan leaves to avoid excessive shading of one another. Removing leaf tips could also reduce shading, but this was advised against in cannabis cuttings by Caplan et al. [11], who found an 18% decrease in rooting success. In mint (*Mentha piperita*) cuttings, Kaymak et al. [41] found no significant effect of cutting lengths (10, 15, 20 cm) on rooting success after inoculation with several PGPR, including *R. rubi* (strain A16). *R. rubi* inoculation provided the highest rooting success (89%) compared to water controls (78%) [41]. In repetition 2 of the current experiment, 100% rooting success was achieved in all treatments and cultivars with an average initial cutting length of 81 ± 1 mm, suggesting a potential threshold length for cannabis cuttings using aeroponic growth units. 

Successful rooting of water-treated cuttings in repetition 2 further suggested that root initiation in vegetative cannabis cuttings will occur under optimal growing conditions. Water-treated cuttings of this experiment required no laboratory facilities or inputs that were incompatible with the official growing framework; therefore, the findings of this work demonstrated a protocol for vegetatively propagating drug-type cannabis in a manner that is suitable for growers in Denmark. This assumes “optimal growing conditions”, which are largely yet to be determined for cannabis grown in indoor, controlled environments [42]. Drawing parallels from hemp production could be biased as it is mostly an outdoor crop grown for its fiber rather than psychoactive compounds [43].

## 4. Materials and Methods

### 4.1. Plant Material

Plant material was obtained from *C. sativa* mother plants of eight different cultivars (Table 2). Seeds were sown in a 50:50 mixture of peat (SW Horto, Hammenhög, Sweden) and vermiculite (Agra-vermiculite, Ommen, The Netherlands). After 30 days, seedlings were transferred to larger pots in the beforementioned substrate mix and placed on a watering table (Drivadan A/S, Søndersø, Denmark). Mother plants were watered from below for 9 min twice a day using an ebb flow system with fertilized water (N-P-K 14-3-23; 150 mg N L^−1^). All plant material was maintained in a greenhouse (located at 55°40′07.8″ N 12°18′28.9″ E) with an 18-h photoperiod and a light intensity of 270 ± 25 µmol m^−2^ s^−1^. Relative humidity (RH) was maintained at 65 ± 10%, and temperature was set to 25 °C. The level of CO_2_ was equal to ambient levels of ca. 419 parts per million (equal to μL L^−1^) globally [44]. Beneficial insects, *Neoseiulus californicus* and *Amblyseius swirskii* (Koppert, Berkel en Rodenrijs, The Netherlands), were released on mother plants every 14 days. 

Mother plants were trimmed by hand with pruning shears seven days prior to taking the cuttings. Cuttings were taken at a length of 6–12 cm cut at a 45-degree angle with at least three fully expanded leaves and 1–2 internodes. Some cuttings required further trimming along the stems to be able to rest on the bottom of 1.5 mL Eppendorf tubes. Cuttings for repetition 1 and repetition 2 were harvested 143 days (3 March 2021) and 185 days (14 April 2021) after sowing mother plants, respectively.

### 4.2. Experimental Setup

In this experiment, cuttings were propagated using three X-Stream Aeroponic growth units (Nutriculture UK Ltd., Skelmersdale, UK). Prior to being maintained in the aeroponic growth units, cuttings were subjected to three treatments consisting of inoculation with *R. rhizogenes* (A4), exposure to IBA, and treatment with water (H_2_O) (Figure 5). 

For treatment A4, an *R. rhizogenes* (A4 strain [21]) suspension was made from an initial bacterial cryostock (stored at −80 °C) mixed with 10 mL Luria–Bertani (LB) medium and placing it on an agitating table at 200 rpm for 18 h at 28 °C. An amount of 10 mL of this suspension was transferred to 90 mL fresh LB media and incubated for another 18 h under identical conditions in order to maintain the culture in an exponential growth phase. The bacterial suspension was adjusted with additional LB medium to OD600 ≤ 0.6, and acetosyringone (Sigma-Aldrich, St. Louis, MO, USA) was added to a final concentration of 15 mg L^−1^. The suspension was incubated for 4 h in the above-mentioned conditions. An amount of 1 mL bacterial suspension was pipetted to 1.5 mL Eppendorf tubes and centrifuged (Centrifuge 5417 R from Eppendorf, Hamburg, Germany) at 1000 rpm for 15 min. The supernatant was replaced with liquid MS medium [45] to avoid exposing the cuttings to the salinity of LB media. For the IBA treatment, a 100 mg L^−1^ IBA (Sigma-Aldrich, St. Louis, MO, USA) solution was prepared based on recommendations for herbaceous cuttings by Kroin [37], and 1 mL was transferred into 1.5 mL Eppendorf tubes. Lastly, for the H_2_O treatment, 1 mL MiliQ water was pipetted into 1.5 mL Eppendorf tubes. 

The bases of cuttings were immersed for 20 h in their respective 1.5 mL Eppendorf tubes prepared above, at 25 °C in darkness, with the purpose of the inoculation (A4), exposure (IBA), or water treatment (H_2_O). Duration of the immersion step RH was maintained at approximately 100% by covering the cuttings with black plastic during immersion. For each repetition, 96 cuttings were taken from 8 *C. sativa* cultivars (Table 2), with 4 specimens per plant per treatment. After immersion, cuttings were transferred to their respective X-Stream Aeroponic Propagator (Nutriculture). This experiment was terminated after 25 days of continuous misting. The nutrient solution that was used for misting consisted of 40 L of tap water (EC ≤ 0.7 mS cm^−1^; pH ≤ 6.5) along with 207 g fertilizer (premixed Pioner NPK(Mg) Makro Blå + Pioner Mikro med jern, both from Azelis Denmark A/S, Kongens Lyngby, Denmark), and 400 mL nitric acid (68%) with final-target values of EC ≤ 1.2 mS cm^−1^ and pH ≤ 6.0; 75 mg N L^−1^ (Table 3). A TinyTag Ultra 2 (Chichester, UK) data logger was placed inside each aeroponic humidity dome and checked weekly with the aim of monitoring greenhouse temperature and RH. 

### 4.3. Root Biometrics

Upon termination of this experiment, cuttings were harvested in succession over three days, one treatment group per day. The order of harvesting was reversed between repetitions. For each cutting, roots, and shoots were separated with scissors, and shoot lengths were measured with a ruler. Roots were photographed using an Epson Perfection V700 scanner (Nagano, Japan). Root scan images were analyzed using WinRhizo^TM^ (Regent Instruments Inc., Quebéc City, QC, Canada) software, which provides biometric data outputs based on images. The parameters of root length and root diameter were obtained using WinRhizo^TM^. The roots and shoots of each cutting were then dried at 70 °C for 72 h. Dry weights of roots and shoots were recorded on a precision scale. In repetition 2, the lengths of cuttings were also measured before the immersion step, and the difference in shoot lengths provided a measure of shoot growth during this experiment. Lastly, the dry weights of roots and shoots were divided to obtain a root/shoot dry weight ratio. In summary, the following seven biometric parameters were recorded: shoot length; shoot dry weight; root/shoot dry weight ratio; root length; root dry weight; root diameter; and shoot growth.

### 4.4. DNA Extraction and PCR

Root samples, ca. 100 mg fresh weight, were collected at the end of each repetition during the processing of cuttings. Root tissue was cut off with scissors from the base of three roots, which was stored at −20 °C. DNA was extracted with the “Plant DNA Isolation Reagent” (TaKaRa Bio Inc., Shiga, Japan) following the manufacturer’s protocol. The concentration of extracted DNA was measured on a Nanodrop™ 1000 by Thermo Fisher Scientific Inc. (Waltham, MA, USA) and adjusted to 10 ng µL^−1^ with MiliQ water. The genes of interest were fragments of *rol*B [46], *vir*D [47], and *EF1α* [48] (Table 4). The *rol*B was included to verify T_L_-DNA insertion, while *vir*D was located outside the T-DNA; its presence can, therefore, indicate *R. rhizogenes* remnants in a sample rather than successful chimeric transformation of a cutting. To assess that the extracted DNA was of sufficient quality for PCR amplification, a reference gene for *C. sativa* was included. Elongation factor-1 alpha (*EF1α*) was chosen based on the work of Guo et al. [48]. PCR components were mixed, and the manufacturer’s instructions for ExTaq polymerase (TaKaRa Bio Inc.) were followed with a final reaction volume of 25 µL and 50 ng of DNA template per reaction. The amplification program for detecting *rol*B, *vir*D, and *EF1α* was 94 °C for 10 min followed by 35 cycles of [94 °C for 30 s, 59 °C for 30 s, 72 °C for 25 s] and a final elongation at 72 °C for 7 min. This was run on a Mastercycler^®^ Pro S (Eppendorf, Hamburg, Germany). Gel electrophoresis was carried out to visualize PCR products on TAE 1.5% agarose gels using a Power PAC 200 (Bio-Rad Laboratories) as a current source.

### 4.5. Statistical Analysis

The rooting performance of eight drug-type cannabis cultivars was analyzed under three treatment groups in two repetitions. In order to determine how biometric data (Section 4.3) across cultivars and repetitions could be combined appropriately, a series of Chi-squared tests were performed on the observed frequencies of rooting success or failure. Additionally, an analysis of variance was carried out in GraphPad version 8.6.4 (San Diego, CA, USA) using one-way ANOVA, followed by Tukey’s HSD test comparing means between treatment groups (*p* ≤ 0.05) for all seven parameters.

## 5. Conclusions

Rooting performance was best in IBA-treated cuttings with increased root length, root dry weight, and root/shoot dry weight ratio in cuttings compared to those treated with *R. rhizogenes* or water. However, IBA inhibited shoot growth, which could be a problem later during commercial production establishment. Superior shoot performance was obtained in *R. rhizogenes* (A4) and H_2_O-treated cuttings, thus showing advantages for the later establishment of commercial production of cannabis. These results are preliminary as a deeper analysis could be obtained on WinRhizo. A variation in cultivar susceptibility toward *R. rhizogenes* was noted in 38% of inoculated cuttings that were successfully infected. The aeroponic growth unit was successful in generating rooted cutting in all tested treatments; therefore, it is an alternative to cannabis producers.

## Figures and Tables

**Figure 1 plants-12-02216-f001:**
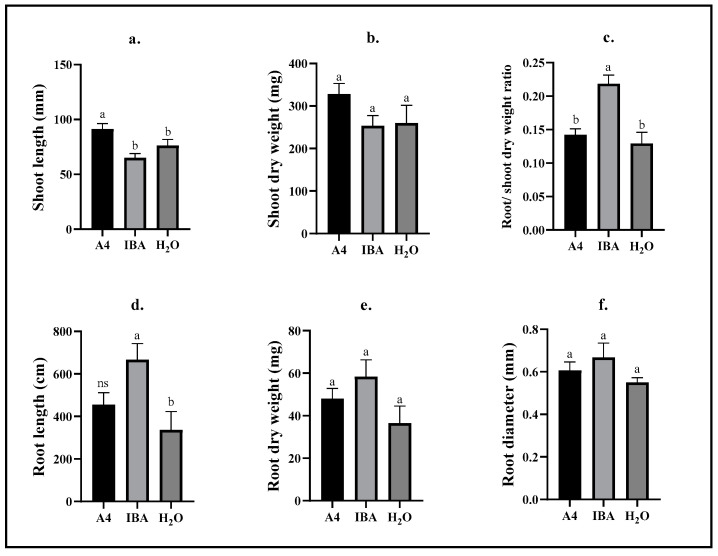
Biometric measures of *C. sativa* cultivars compiled in the first repetition of this experiment. Cuttings of *C. sativa* cultivars “Hindu Kush”, “California Orange”, “Herijuana”, “The Pure”, “Bruce Banner”, “Big Bud”, “Motherlode Kush”, and “Wild Thailand” were maintained in aeroponic growth units for 25 days following a 20 h immersion treatment in which cuttings were either inoculated with *R. rhizogenes* (A4), exposed to IBA (IBA) or treated with water (H_2_O). (**a**) The length of shoots measured from the bottom to the youngest point of growth (mm). (**b**) The dry weight of shoots was recorded after 70 h of drying in a 70 °C drying chamber (mg). (**c**) Root dry weight was divided by shoot dry weight to estimate growth ratios. (**d**) The root length as estimated by analyzing root scan images in WinRhizo^TM^ (cm). (**e**) Roots were dried for 70 h at 70 °C in a drying chamber and subsequently weighed (mg). (**f**) The root diameter was estimated by analyzing root scan images in WinRhizo^TM^ (mm). Different letters between columns in a graph indicate a statistically significant difference (*p* ≤ 0.05) between treatments using Tukey’s HSD test; ns indicates no significant difference compared to any other treatment group. Bars represent means ± standard error; *n* = 23–32.

**Figure 2 plants-12-02216-f002:**
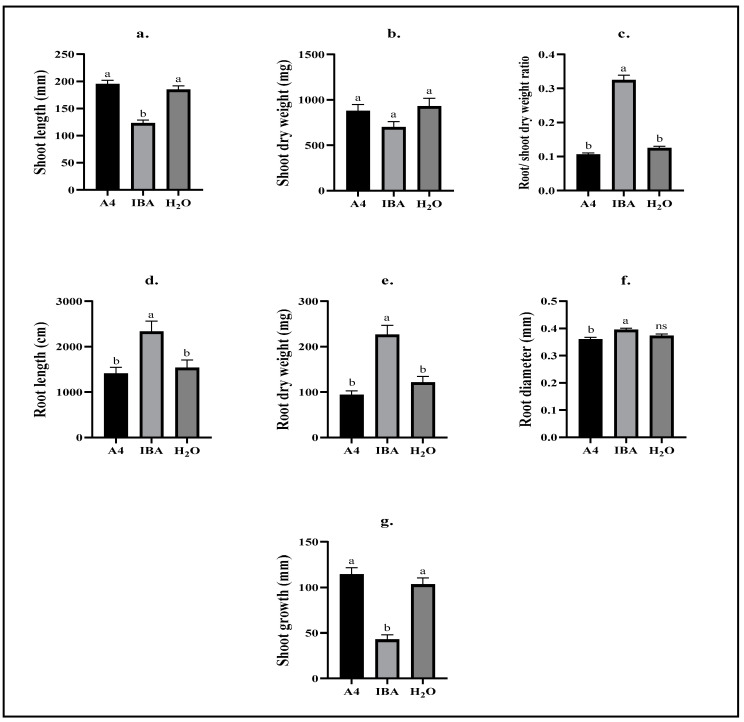
Biometric measures of *C. sativa* cultivars compiled in the second repetition of this experiment. Cuttings of *C. sativa* cultivars “Hindu Kush”, “California Orange”, “Herijuana”, “The Pure”, “Bruce Banner”, “Big Bud”, “Motherlode Kush”, and “Wild Thailand” were maintained in aeroponic growth units for 25 days following a 20 h immersion treatment, in which cuttings were either inoculated with *R. rhizogenes* (A4), exposed to IBA (IBA), or treated with water (H_2_O). (**a**) The length of shoots measured from the bottom to the youngest point of growth (mm). (**b**) The dry weight of shoots was recorded after 70 h of drying in a 70 °C drying chamber (mg). (**c**) Root dry weight was divided by shoot dry weight to estimate growth ratios. (**d**) The root length was estimated by analyzing root scan images in WinRhizo^TM^ (cm). (**e**) Roots were dried for 70 h at 70 °C in a drying chamber and subsequently weighed (mg). (**f**) The root diameter was estimated by analyzing root scan images in WinRhizo^TM^ (mm). (**g**) Shoot growth (mm) Different letters between columns in a graph indicate a statistically significant difference (*p* ≤ 0.05) between treatments using Tukey’s HSD test; ns indicates no significant difference compared to any other treatment group. Bars represent means ± standard error; *n* = 23–32.

**Figure 3 plants-12-02216-f003:**
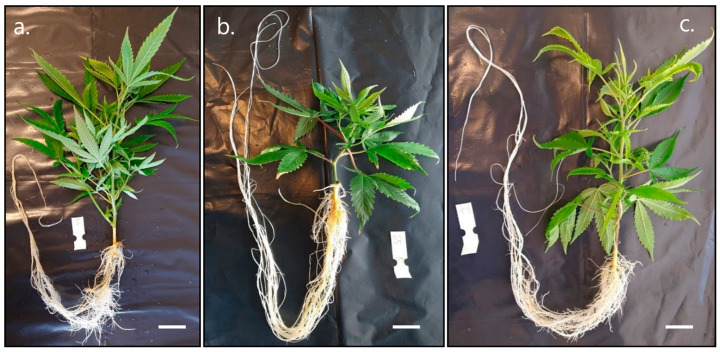
Visualization of different root/shoot ratios in *C. sativa* cuttings. “Hindu Kush” cuttings from each treatment during harvest and processing of repetition 2. (**a**) Cutting treated with *R. rhizogenes* A4. (**b**) Cutting treated with IBA. (**c**) Cutting treated with H_2_O with shoot lengths measuring 22.6, 11.5, and 20.5 cm, respectively. Bar ≤ 2 cm.

**Figure 4 plants-12-02216-f004:**
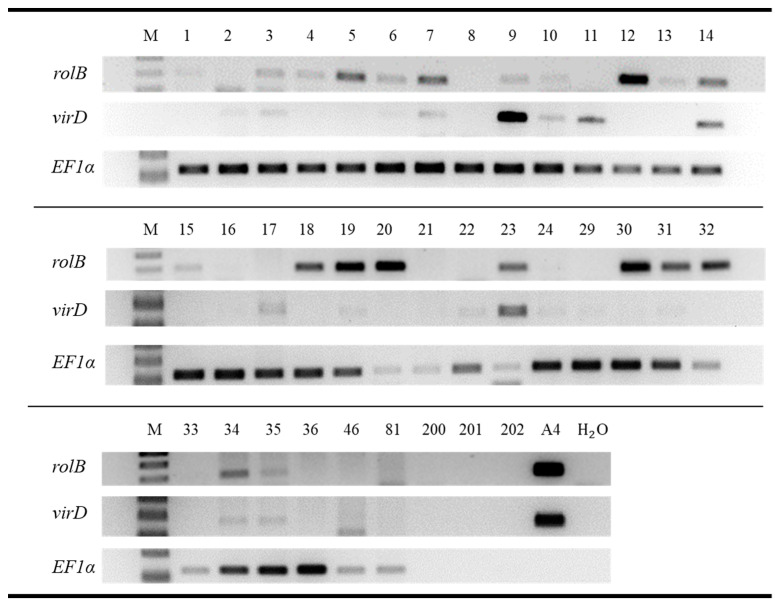
Presence of *rol*B, virD, and EF1α fragments in *C. sativa* root tissue, nutrient solutions of aeroponic growth units, and purified water. Cuttings of *C. sativa* cultivars “Hindu Kush”, “California Orange”, “Herijuana”, “The Pure”, “Bruce Banner”, “Big Bud”, “Motherlode Kush”, and “Wild Thailand” were either inoculated with *R. rhizogenes* (A4), exposed to IBA (IBA), or treated with water (H_2_O) for 20 h and maintained in aeroponic growth units for 25 days. Root tissue samples were taken upon experimental termination. PCR products were visualized by 1.5% agarose gel electrophoresis. M: 100 bp ladder; 1–36: A4 cuttings; 46: IBA cutting; 81: H_2_O cutting; 200, 201, 202: nutrient solution samples collected from A4, IBA, and H_2_O treatment groups, respectively; A4: positive control; H_2_O#1: Negative control.

**Figure 5 plants-12-02216-f005:**
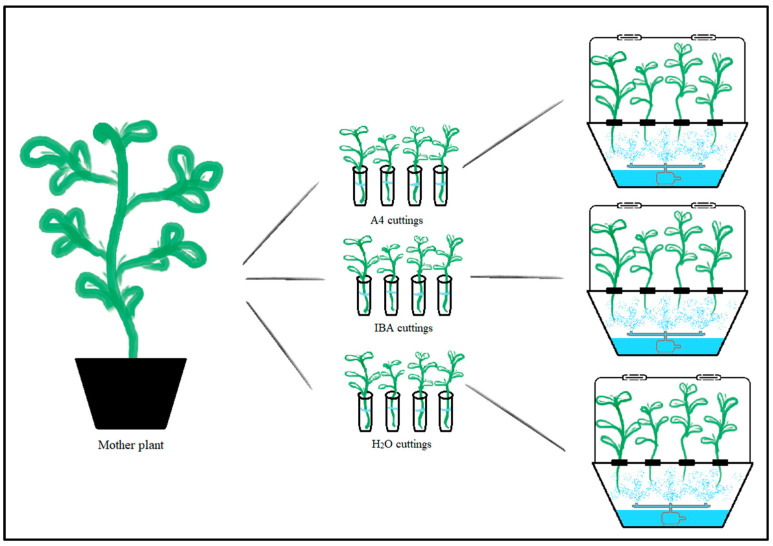
Illustration of the experimental setup. A mother plant from each *C. sativa* cultivar (**left**) provided 12 cuttings, which were separated into three groups (**middle**) according to their respective immersion treatment: A4 inoculation; IBA exposure; or H_2_O treatment. Lastly, in the (**right**) part, there is the aeroponic system and cutting representation. Cuttings were immersed for 20 h in their respective treatment and maintained in aeroponic growth units for 25 days.

**Table 1 plants-12-02216-t001:** Rooting success of *C. sativa* cuttings listed by treatment and cultivar for both repetitions. The treatments included inoculation with *R. rhizogenes* (A4), dipping in IBA 100 mg L^−1^ solution or simply deionized water for 24 h prior to transferring to the aeroponics systems. A cutting was considered successfully rooted if visible root(s) had emerged after 25 days of aeroponic maintenance. In the right column, average rooting successes (%) across both repetitions are listed as mean ± standard error (SE).

Treatment	Cultivar	Repetition 1			Repetition 2			Average
		N. of Cuttings	Rooted Cuttings	Rooting Success, %	N. of Cuttings	Rooted Cuttings	Rooting Success, %	Rooting Success ± SE, %
A4	“Hindu Kush”	4	3	75	4	4	100	95 ± 5
	“California Orange”	4	4	100	4	4	100	
	“Herijuana”	4	3	75	4	4	100	
	“The Pure”	4	4	100	4	4	100	
	“Bruce Banner”	4	3	75	4	4	100	
	“Big Bud”	4	4	100	4	4	100	
	“Motherlode Kush”	4	4	100	4	4	100	
	“Wild Thailand”	4	4	100	4	4	100	
IBA	“Hindu Kush”	4	4	100	4	4	100	98 ± 3
	“California Orange”	4	4	100	4	4	100	
	“Herijuana”	4	4	100	4	4	100	
	“The Pure”	4	4	100	4	4	100	
	“Bruce Banner”	4	3	75	4	4	100	
	“Big Bud”	4	4	100	4	4	100	
	“Motherlode Kush”	4	4	100	4	4	100	
	“Wild Thailand”	4	4	100	4	4	100	
H_2_O	“Hindu Kush”	4	3	75	4	4	100	86 ± 14
	“California Orange”	4	4	100	4	4	100	
	“Herijuana”	4	4	100	4	4	100	
	“The Pure”	4	3	75	4	4	100	
	“Bruce Banner”	4	1	25	4	4	100	
	“Big Bud”	4	4	100	4	4	100	
	“Motherlode Kush”	4	0	0	4	4	100	
	“Wild Thailand”	4	4	100	4	4	100	
	Total	96	83	86	96	96	100	

**Table 2 plants-12-02216-t002:** List of *C. sativa* cultivars included in this experiment.

Cultivar	Breeder	Breeder Location
“Hindu Kush”	Sensi Seeds	Amsterdam, The Netherlands
“California Orange”	Seedsman	Barcelona, Spain
“Herijuana”	Sannie’s Seeds	Netherlands
“The Pure”	Flying Dutchmen	Netherlands
“Bruce Banner”	N/A *	N/A *
“Big Bud”	Sensi Seeds	Amsterdam, The Netherlands
“Motherlode Kush”	Sannie’s Seeds	The Netherlands
“Wild Thailand”	World of Seeds	Spain

* N/A—not available.

**Table 3 plants-12-02216-t003:** Initial concentration of macro- and micronutrients in each aeroponic propagator. The contents of each aeroponic propagator were 40 L of tap water (EC ≤ 0.7 mS cm^−1^; pH ≤ 6.5), 207 g premixed fertilizer yielding the concentrations below, and 400 mL nitric acid (68%). Final EC ≤ 1.2 mS cm^−1^; pH ≤ 6.0; 75 mg N L^−1^.

Nutrient	Concentration (mg L^−1^)
Nitrogen, N	75
Phosphorus, P	16
Potassium, K	120
Magnesium, Mg	16
Boron, B	0.12
Copper, Cu	0.07
Iron, Fe	0.68
Manganese, Mn	0.26
Molybdenum, Mo	0.03
Zinc, Zn	0.09

**Table 4 plants-12-02216-t004:** Target genes (*rol*B, *vir*D) and reference gene (*EF1α*), along with specific primer pair sequences and their fragment lengths.

Gene		Primer Sequences (5′-3′)	Fragment Length (bp)
*rol*B	Forward	GATATCCCGAGGGCATTTTT	182 ^1^
	Reverse	GAATGCTTCATCGCCATTTT	
*vir*D	Forward	CTGAATTACGACGCCTTGCG	196 ^2^
	Reverse	TGCGATGACGACTGTTCCAA	
*EF1α*	Forward	AGCGTGGTATCACCATTGAC	200 ^3^
	Reverse	AGCACAATCAGCCTGTGAAG	

^1^ [46] ^2^ [47] ^3^ [48].

## Data Availability

Raw data available upon request.

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
