# Peer review of "Alternative Rooting Methods for Medicinal Cannabis Cultivation in Denmark—Preliminary Results"

_plants, 2023, doi:10.3390/plants12112216_

Round 1

Reviewer 1 Report

Dear authors, thank you for your article which is current and interesting. I read your article and I have some questions that might help you:

1: The number of samples for each treatment is low for statistical analysis

2: The two repetitions provide different effects on the control plant. could the number of days affect the rooting process?

3: why did you report only root length and root diameter? Winrhizo can provide many parameters on root morphology that could improve the quality of your work.

Author Response

Dear Reviewer 1

1: The number of samples for each treatment is low for statistical analysis

Thank you for raising this issue, it was a mistake in which we did not consider that that data was merged for the cultivars in both repetitions, hence the correct n is 23-32. This has been corrected in the manuscript in figure 1 and figure 2 legends. Sorry for the confusion.

2: The two repetitions provide different effects on the control plant. could the number of days affect the rooting process?

Thank you for this comment and suggestion. We observed variation between repetitions across all measured biometrics, most of which measured higher in the second repetition. We suspected that minor inconsistencies in the methodological workflow in combination contributed to differences in rooting success. We also speculated whether rooting success could be related to the overall state of the mother plants, being older, but we do not have data from this experiment to support such a claim. We also reckon that the number of days could have an effect.

3: why did you report only root length and root diameter? Winrhizo can provide many parameters on root morphology that could improve the quality of your work.

Thank you for this question. We acknowledge that only a few parameters that WinRhizo can provide were included in this work. Experienced users of WinRhizo at our faculty were involved in order to choose parameters that best addressed the purpose of this work, namely rooting success and biomass allocation, which then became our focus.

For future work, it would be interesting to record a wider range of parameters and describe the rooting morphology of chimeric and eventually transformed and regenerated cannabis cuttings in detail.

Reviewer 2 Report

Comments on the review manuscript id: plants-2390114 entitled "Alternative rooting methods for medicinal cannabis cultivation in Denmark"

In my opinion, the objectives set by the Authors of the paper have been achieved. The paper is interesting, well written and logical. It also has a practical and innovative aspect.  I have highlighted minor comments in the text regarding minor textual errors and a question regarding the presentation of the results of the statistical analyses. 

Author Response

Thank you very much for your valuable comments

Reviewer 3 Report

A manuscript titled "Alternative Rooting Methods for Growing Medical Cannabis in Denmark" seems like a good document.

     The results could be useful for future rooting experiments in the same or another species. Therefore, in my opinion, the manuscript is suitable for publication after minor improvements.

The Introduction does not provide sufficient background and needs to be improved in the part dealing with alternative substances to IBA, taking into consideration for example biostimulants (algae extracts, protein hydrolysates etc.) which have recently been used in propagation by cuttings in cannabis or in other species such as shrubs for example. The Material and Method section is clear and presents the experimental factors clearly. The results are presented very clearly. The discussion and conclusions are well written. the introduction of academic articles relating to other alternative substances to the IBA used in the multiplication by cuttings is recommended.

Best regards.

Author Response

Dear Reviewer 3

A manuscript titled "Alternative Rooting Methods for Growing Medical Cannabis in Denmark" seems like a good document.

The results could be useful for future rooting experiments in the same or another species. Therefore, in my opinion, the manuscript is suitable for publication after minor improvements.

Thank you for the positive comments.

The Introduction does not provide sufficient background and needs to be improved in the part dealing with alternative substances to IBA, taking into consideration for example biostimulants (algae extracts, protein hydrolysates etc.) which have recently been used in propagation by cuttings in cannabis or in other species such as shrubs for example.

We follow this issue and have conducted a further literature study, which did not provide further scientific data. However, our cannabis grower contact confirms the wider user of algae extracts and protein hydrolysates in cannabis production. We have now included a sentence describing this broadly – to address this issue as well – as we see the importance.

The Material and Method section is clear and presents the experimental factors clearly. The results are presented very clearly. The discussion and conclusions are well written. the introduction of academic articles relating to other alternative substances to the IBA used in the multiplication by cuttings is recommended.

We have now made updates accordingly in that part.

Round 2

Reviewer 1 Report

dear Authors, thanks for your correction. I suggest to insert in the title and in the manusctipt the terms "preliminary results" since, for example for winrhizo, you didn't show all the results. 

Author Response

Dear Reviewer 1, thank you for your second round of comments and we will of course accommodate your suggestion to include preliminary results in the title and manuscript (see new file).